# Into the spotlight: A spatial study of potentially underreported leptospirosis among dengue-negative patients in São Paulo city, Brazil

Stephanie Bergmann Esteves[1,2]*, Luciano Marcondes de Oliveira[1,3], Aline Gil Alves Guilloux[1], Adriana Cortez[4], Eduardo de Masi[3], Isabelle Martins Ribeiro Ferreira[3,5], Evelyn Moura de Lima[2], Gabriele Yumi Ramalho[2], Priscila de Castilho Luna[4], Jose Soares Ferreira Neto[1], Marcos Bryan Heinemann[1], Bruno Alonso Miotto[4]

**1** Departamento de Medicina Veterinária Preventiva e Saúde Animal, Faculdade de Medicina Veterinária e Zootecnia da Universidade de São Paulo, São Paulo, Brazil, **2** Universidade Santo Amaro, Faculdade de Medicina Veterinária, São Paulo, Brazil, **3** Secretaria Municipal da Saúde de São Paulo, Divisão de Vigilância de Zoonoses, São Paulo, Brazil, **4** Universidade Santo Amaro, Programa de Pós-graduação em Saúde Única, São Paulo, Brazil, **5** Instituto Adolfo Lutz, Centro de Parasitologia e Micologia, São Paulo, Brazil

* sbergmann@usp.br

## Abstract

Leptospirosis and dengue share similar unspecific symptoms, complicating differential diagnosis in endemic regions. This overlap is further exacerbated by the often-underrecognized nature of leptospirosis, resulting in low clinical suspicion among healthcare providers. Understanding the scale of underdiagnosed leptospirosis cases, particularly among dengue-negative patients, is critical for improving public health responses. This cross-sectional study analyzed data from 6,936 febrile patients who tested negative for dengue at public health services across São Paulo city. Serum samples from these patients were subsequently tested for anti-*Leptospira* IgM antibodies. Spatial analysis was conducted to identify areas at increased risk of underdiagnosed leptospirosis, and these findings were compared to cases reported in the Brazilian's Information System for Notifiable Diseases from 2009 to 2019. Our results revealed that, from the 6,936 patients tested, 786 (11.3%) were seroreactive for leptospirosis, with a higher prevalence among women (55.7%; p=0.003). Spatial analysis identified 18 high-risk clusters for potentially unrecognized leptospirosis, predominantly in peripheral regions with socioeconomic vulnerabilities. Notably, two significant high-risk areas were located in the North (RR=2.13) and South (RR=2.69) regions of the city. These findings underscore the urgent need for targeted public health interventions to improve disease surveillance and diagnostic capacity, particularly in the high-risk areas identified. Addressing underrecognition of leptospirosis is essential for reducing morbidity and mortality associated with the disease. Future research should expand on this work by integrating broader temporal, environmental, and socioeconomic data.

**Data availability statement:** All relevant data are within the manuscript and its Supporting Information files.

**Funding:** This study received funding support in the form of fellowships granted to the authors. SBE was supported by the São Paulo Research Foundation (FAPESP, grant number 2021/02534-9, https://fapesp.br/). MBH was supported by the National Council for Scientific and Technological Development (CNPq, grant number 310462/2021, https://www.gov.br/cnpq/). The funders had no role in study design, data collection and analysis, decision to publish, or preparation of the manuscript.

**Competing interests:** The authors have declared that no competing interests exist.

## Author summary

Leptospirosis is a neglected bacterial disease that often affects vulnerable populations in urban areas of developing tropical countries. Due to overlapping clinical symptoms with dengue, many cases of leptospirosis remain undiagnosed, especially among patients who test negative for dengue. This study represents the first large-scale investigation aimed at detecting possibly underrecognized cases of leptospirosis among dengue-negative patients. In São Paulo, Brazil, a centralized system for processing suspected dengue samples enabled this large-scale investigation of underrecognized leptospirosis cases. Among 6,936 patients tested, 786 had anti-Leptospira IgM antibodies, with a higher prevalence observed among women. Spatial analysis identified 18 high-risk clusters for potentially unrecognized leptospirosis, predominantly in peripheral regions with socioeconomic vulnerabilities. Notably, two significant high-risk areas were located in the North and South regions of the city. We hope that these findings should provide a critical foundation for local health authorities to design and implement targeted strategic interventions aimed at reducing the underrecognition of leptospirosis. Such interventions include educational campaigns targeting local populations, field training for family medicine professionals on the importance and risk factors of the disease, as well as raising awareness among primary care teams about the high number of underdiagnosed leptospirosis cases. Additionally, providing instructions to local healthcare workers on clinical and laboratory differentiation between leptospirosis and dengue based on previous studies that have identified comprehensive screening schemes and algorithms should be also considered. Furthermore, it is hoped that the design of this study may serve as a model for assessing risk areas for unreported leptospirosis in other locations that share the same epidemiological and socioeconomic features found in São Paulo city.

## Introduction

Leptospirosis is a disease that affects both humans and animals globally, caused by spirochetal pathogenic bacteria of the genus *Leptospira* [1]. Transmission primarily occurs through contact with soil and water sources contaminated by the urine of infected animals, particularly synanthropic and domestic mammals [2]. Recent evidence, however, has revealed novel microbiological aspects involved in the transmission of the disease, such as the prolonged environmental survival of pathogenic *Leptospira* in highly diversified field conditions [3–5], and the apparent ability of the bacteria to replicate outside mammalian hosts [6], highlighting the need for a broader understanding of the disease within a One Health framework.

High levels of rainfall, humidity, and temperature are well-established environmental drivers of the disease [2], especially in tropical regions, where morbidity is typically higher in impoverished populations living under low sanitation levels and intense interactions with animals [7,8]. In urban settings, the disease is strongly associated with flooding and proximity to open sewage [9–11]. Urban disadvantaged communities, which are highly vulnerable to natural disasters and extreme weather events, are particularly at risk, mainly for presenting precarious housing conditions, poor sanitation, overpopulation, and constant rodent infestation [12]. Furthermore, inadequate healthcare provision by local authorities, failure to implement or improve sanitation infrastructure, lack of garbage removal, and inefficient policies targeting the reduction of poverty perpetuate a cycle of neglect, further increasing the disease burden in these communities [13,14].

Due to these characteristics, leptospirosis is considered by many scholars to be a neglected zoonosis [9,15], though it is not currently included in the WHO list of tropical neglected diseases, which – like leptospirosis – are often associated with severe health, social, and economic impacts. Yet, it is estimated that more than one million cases of leptospirosis occur annually worldwide, leading to nearly 60,000 deaths and the loss of approximately 2.9 million disability-adjusted life-years (DALYs) per year [8,16]. These estimates reveal that leptospirosis rank among the leading zoonotic causes of morbidity and mortality in humans [16], and since it is largely under-recognized [17], the actual disease burden is likely to be much higher than reported.

Surveillance data shows that more than 60% of leptospirosis alerts occur in the Americas [18], with Latin America and the Caribbean reporting the majority of the disease outbreaks [19]. Approximately 10,000 human cases are registered annually in Latin America alone, with Brazil accounting for 40% of these cases and 90% of the region's reported deaths [20,21]. In Brazil approximately 4,000 cases are registered annually, with higher numbers in urban areas [22]. Most cases are reported in the state of São Paulo, particularly in São Paulo city [23,24], the largest city in southern hemisphere, with over 12 million habitants. Despite generating more than 10% of the national GDP and being the largest financial and commercial hub in the country, São Paulo continues to face stark socioeconomic inequalities, with approximately 2 million families living in poverty and an additional 800,000 in extreme poverty [25], most of whom reside in areas with a high risk of leptospiral transmission.

Estimating the impact of leptospirosis in São Paulo – as in any other location –, remains challenging, though. Despite the large number of people living at risk, the number of confirmed cases of leptospirosis in the city of São Paulo has remained low in recent years, with about one to two hundred confirmed cases per year [26]. It is a consensus that the surveillance system has not been effective to identify and notify the oligosymptomatic and mild cases. Only severe cases, of which many need to be hospitalized, are suspected. Indeed, the disease most often presents with mild and non-specific clinical manifestations, similar to numerous other febrile illnesses, such as dengue and malaria [27]. Several studies worldwide have identified leptospirosis as an important etiological pathogen in patients presenting with acute fever of unknown origin [27–29], and a high proportion of undetected cases of leptospirosis among febrile patients is expected.

Laboratory testing is usually required to confirm acute leptospiral infection, and methods for direct detection, such as culture and nucleic acid amplification are extremely helpful during the onset of the disease [1], whereas serological methods, such as the microscopic agglutination test (MAT), reagent enzyme-linked immunosorbent assay (ELISA) and rapid lateral flow immunochromatography assays are strongly recommended to confirm infection during the later stages of acute infection and convalescent phase [30,31].

ELISA tests can typically detect IgM antibodies against leptospires as early as 4–5 days after clinical symptoms appear [1], making it more suitable for the early detection of the disease than MAT. Moreover, the assay is user-friendly, requiring minimal training and providing results within 2 to 4 hours [1], unlike the MAT, which involves tedious, labor-intensive procedures and requires well-equipped laboratories with highly experienced staff. Considering these differences, it is unsurprising that up to a third of spatial epidemiological studies on leptospirosis described in the literature rely on detection through ELISA testing [32], following the World Health Organization (WHO) endorsement of IgM ELISA for the serodiagnosis of leptospirosis in areas with limited healthcare resources [33].

Although multiple diagnostic tests are available for use, the overall lack of suspicion at clinical presentation and the often-poor awareness of the disease's epidemiology among at-risk populations, and even among healthcare professionals, remain key determinants for disease

underreporting, especially in areas where the high incidence of dengue often leads to an overestimated suspicion of dengue infection, potentially overshadowing the occurrence of leptospirosis.

Dengue, a mosquito-borne infection, is a major public health threat in most of the developing world, particularly in the Americas. In 2023, more than 4.5 million confirmed cases of dengue were reported in the Americas [34]. In Brazil, the country with the highest number of cases in the Americas, 2.9 million patients were officially registered as dengue cases in 2023 [35], with São Paulo being the Brazilian state with the second-highest number of reported cases [36]. Despite these staggering numbers, most patients with suspected dengue fever do not have a definitive diagnosis for the viral infection, demonstrating that the etiology of the febrile illness, in most cases, remains undetermined. In São Paulo city, the proportion of negative cases for dengue among patients initially suspected of dengue is strikingly high. In 2023, for example, 62,997 suspected cases attended at public facilities had samples tested by serological methods for confirmation of infection, of which only 13,716 (21.8%) were actually confirmed [37].

Given the similarities of symptoms shared between patients with dengue and leptospirosis, many authors have investigated the presence of leptospiral infection among febrile patients initially suspected of having dengue, revealing high proportions of leptospiral infection among dengue-negative populations [38–41]. This approach has successfully guided clinical and diagnostic decisions to promote timely identification of leptospirosis cases, which is essential to reduce lethality [42], also helping local surveillance workforce to assess and scale the disease underestimation.

Despite their relevance, these studies are mostly conducted in small-scale hospital settings with limited sample sizes, which restricts a broader understanding of the factors contributing to the disease's underrecognition, also preventing the identification of high-risk areas for underdiagnosis. Interestingly, in São Paulo city, nearly all samples from suspected dengue patients attending at public health services are processed at a single facility. This centralized pipeline enables large-scale investigations into unreported cases of leptospirosis among dengue-negative patients, allowing further exploration into the spatial patterns of such cases.

The present study describes the identification of potentially unrecognized cases of leptospirosis among dengue-negative patients presenting with febrile illness of unknown etiology at public health services in all regions of São Paulo city. Further spatial analysis was carried out to determine risk areas for the underdiagnosis of the disease to assist local health authorities in improving preventive strategies against human leptospirosis.

## Methods

### Ethics Statement

The present study was approved by the Bioethics Committee of the São Paulo Municipal Health Department (Plataforma Brasil CAAE 36678820.0.0000.0086). Given that this study utilized exclusively anonymized, retrospective data, the requirement for formal consent was waived in accordance with local regulations on personal data protection. All procedures and data analyses were conducted in full compliance with these regulatory standards.

### Study design and participants

This is a cross-sectional study that assessed the presence of IgM antibodies against leptospires in dengue-negative febrile patients treated at the public health services from São Paulo city. Blood samples were taken between May and December of 2019 from patients presenting fever associated with myalgia, headache, rashes, or blood disorders for more than four days, who sought medical care in one of the 602 primary healthcare units located across the city (Fig 1).

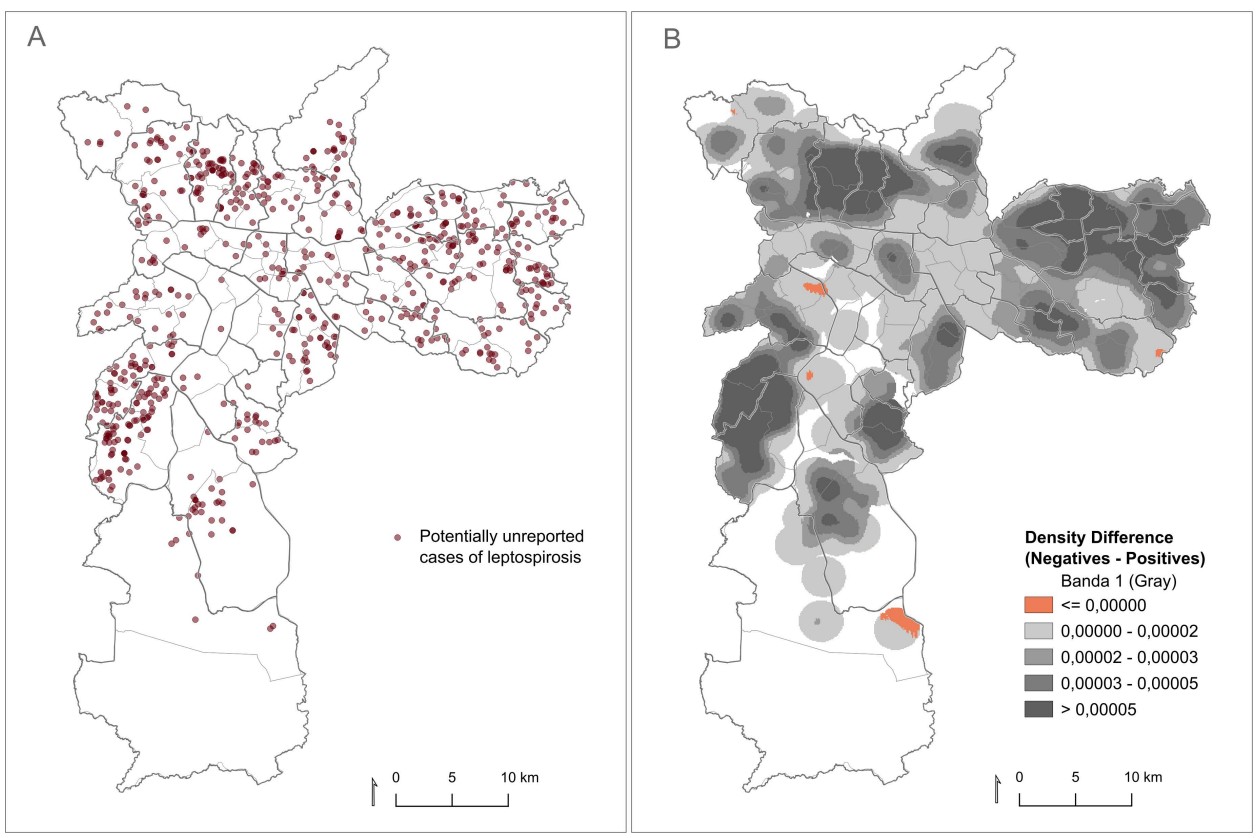

**Fig 1. (A) Spatial distribution of potentially unreported cases of leptospirosis; and (B) density difference map considering kernel estimate of controls with the density of cases subtracted. The maps were built using the free and open source QGIS software** (https://www.qgis.org/en/site/) **based on shapefiles obtained from Instituto Brasileiro de Geografia e Estatística (IBGE -** https://www.ibge.gov.br/geociencias/organizacao-do-territorio/malhas-territoriais/26565-malhas-de-setores-censitarios-divisoes-intramunicipais.html**).**

All subjects initially had clinical suspicion of dengue and had serum samples sent to the Zoonoses and Vector-Borne Diseases Diagnostic Laboratory (LABZOO/DVZ) – a public facility belonging to the municipal Health Surveillance Coordination (COVISA) of the Brazilian Unified Health System (SUS) – for testing against dengue infection using the Panbio Dengue IgM Capture ELISA (Abbott, USA). Only patients showing negative results for dengue infection had serum samples eligible for further leptospiral testing.

A total of 12,988 subjects were tested for dengue infection during the studied period, of which 7,064 presented negative results. Of these, 81 samples were excluded due to incorrect identification, and another 47 for having insufficient volume to perform additional tests, totaling 6,936 samples included to test for leptospiral infection.

The presence of anti-*Leptospira* antibodies was assessed using the commercial Panbio Leptospira IgM ELISA kit (Abbott, USA) according to the manufacturer's instructions. Seroreactivity was considered the main outcome in this study, and patients with positive reaction for the ELISA detection were considered probable underreported cases of leptospirosis (cases), whereas non-reactive patients were considered as being not infected by leptospires (controls).

All ELISA-positive patients were further tested using the Microscopic Agglutination Test (MAT) with a panel of 24 serogroups, as recommended by the WHO. Patients were considered MAT-positive if they presented more than 50% agglutination at a dilution of 1:100, in accordance with the standard MAT protocol.

All patients (cases and controls) were registered as suspected cases of dengue at presentation in the Information System for Notifiable Diseases (SINAN in Portuguese) – a comprehensive database used by the Ministry of Health to track and monitor the incidence of diseases that are legally required to be reported by healthcare providers in Brazil – which allowed assessment of socio-demographic information and approximate residence location for further descriptive, spatial, and risk analyses. São Paulo's environmental and populational data were retrieved from public databases and spatially joined with patients' individual information to evaluate associations with the seropositivity outcome. The patients' residence addresses were also used to produce a heat map for exploratory spatial analysis.

For the identification of areas with increased risk for under-recognition of leptospirosis, a territorial approach using two different techniques was carried out: (i) the first estimated the seropositivity rate according to the UDH (Human Development Units, from the acronym in Portuguese), which was the smallest cartographical unit available; and (ii) a spatial cluster analysis was performed, considering the individual locations of cases and controls over the population density within each UDH. To compare the distribution of potentially underreported cases of leptospirosis, we included the official cases of leptospirosis reported by SINAN in the city of São Paulo between 2009 and 2019.

## Data sources and variables

Information on patients' addresses, date of birth, gender, level of education, date of sample collection, and dengue or leptospirosis test results were extracted from the SINAN registration form, which was made available upon request to the local health surveillance authorities.

Environmental and urbanistic variables tested for association with seropositivity outcome were retrieved from public cartographic repositories for the municipality of São Paulo, available on the DataGeo (http://datageo.ambiente.sp.gov.br/), IBGE (https://www.ibge.gov.br/geociencias/todos-os-produtos-geociencias.html) and GeoSampa (https://geosampa.prefeitura.sp.gov.br/) websites, and from the Brazilian Human Development Atlas (https://dados.gov.br/). Based on the cartographic data, which were available in shapefile formats, the following variables were determined: whether patients were living in urban or rural areas; slum areas; slums undergoing an urbanization process; irregular settlements; forest conservation units; and areas at risk of flooding. We also incorporated data on the locations of public healthcare facilities responsible for referring samples to LABZOO. The distribution of cases and controls was reassessed using the UDH territorial subdivision to allow further investigation into the possible influence of the Municipal Human Development Index (IDHM) and its dimensions – Income, Education, and Longevity – on the seropositivity outcome.

## Statistical analysis and risk assessment

The patients' geographic location was retrieved from SINAN using latitude and longitude coordinates in decimal format. For patients with only zip codes available, the Google Maps service was used to calculate a random point within the polygon of the postal address code provided. The number of decimal coordinates was reduced from 5 to 3, decreasing the precision and preventing the retrieval of the patient address through reverse geocoding. The applied error was approximately 50 meters, both for latitude and longitude. The coordinates of patients' addresses were further used to assess the distribution of cases and controls across the territory, including the assessment according to the UDH subdivision.

Cartographic data regarding the patients' location were transferred to the QGIS 3.14.0 software for spatial visualization. The location and data were plotted on the map and spatially joined with publicly available shapefiles describing slum areas, slums undergoing the

urbanization process, irregular settlements, forest conservation units, areas at risk of flooding, urban or rural areas, public primary healthcare centers, UDH units, and the IDHM to create the final database.

For the patients' level analysis, the association between the frequency of cases and controls and possible qualitative risk factor variables was tested using the chi-square test; the association between the frequency of cases and controls with quantitative variables was carried out using the Wilcoxon test. For exploratory spatial analysis, two normalized kernel density surfaces using a quartic function were initially calculated, with a radius of 2,000 meters and a grid size of 100x100 pixels for the cases and control points. The density differences were assessed to evaluate the impact of population clusters. In addition to this analysis, density maps for leptospirosis cases were generated. To ensure appropriate comparison of proportions, density maps for both probable underreported leptospirosis cases and officially recorded cases from SINAN were generated using the same configuration parameters and normalization of density values. The difference between these density surfaces was then calculated to identify areas with higher densities of underreporting compared to those of officially identified cases.

The spatial scanning methodology applied for the identification of areas with increased risk for under-recognition of leptospirosis was described by Kulldorff and Nagarwalla [43] with the support of SatScan 9.5 software. For each possible cluster, we implemented the maximum likelihood test for a null hypothesis that the risk of being seropositive is similar between the areas inside and outside the ellipse, compared to the assumption that the risk is greater inside the ellipse. The ellipse with the maximum likelihood value was considered the most likely cluster to indicate risk, while not disregarding possible secondary clusters, due to the possibility of more than one area having environmental conditions favorable to leptospirosis transmission. The significant ellipses were represented graphically using QGIS software [44].

The data were described with frequency and confidence intervals for qualitative variables and with measures of central tendency and dispersion for quantitative data. The tests were conducted considering a two-way α of 0.05 and a 95% confidence interval (95% CI), using the R software [45].

## Results

Out of the 6,936 patients tested, 786 (11.3%; 95% CI 10.6%–12.1%) had detectable IgM antibodies against leptospires, whereas 6,150 patients (88.7%; 95% CI 87.9%–89.4%) had no serological evidence of leptospiral infection. Among these patients, 50.8% (95% CI 49.5%–52%) were women and 49.2% (95% CI 48%–50.5%) were men (Table 1), with a mean age of 32 years (SD ± 17 years).

For the seroreactive patients (cases), the proportion of women was higher (55.7%; 95% CI 52.2%–59.4%), with a statistically significant association (p = 0.003). The mean age in this group was 33 years (SD ± 13.6 years), with no significant difference compared to control patients. There was also no difference in education levels between the case and control subpopulations, with both groups having 39 to 44% of patients with only elementary school education.

The MAT test was performed on 784 samples that tested positive for the ELISA, as 2 samples (0.3%) did not have sufficient volume for diagnostic testing. Among the 784 samples tested, 7 samples (0.9%; 95% CI: 0.4%–1.5%) were positive. Of these, three samples tested positive for the Butembo serovar of the Autumnalis serogroup, with a titer of 200; two samples tested positive for the Castellonis serovar of the Ballum serogroup, with one sample showing a titer of 100 and the other a titer of 200; one sample tested positive for the Icterohaemorrhagiae serovar of the Icterohaemorrhagiae serogroup, with a titer of 100; and one sample tested positive for the Panama serovar of the Panama serogroup, with a titer of 200.

**Table 1. Descriptive statistics of case and control groups using 95% confidence interval (95% CI) for sex and educational level categories; and mean values with standard deviation (± SD) for the age category. P-values for group comparisons with chi-square test are also shown.**

| | Total | | Positive | | Negative | | p-value |
|---|---|---|---|---|---|---|---|
| | N | % | N | % (95%CI) | N | % (95%CI) | |
| **Sex** | | | | | | | |
| Female | 3522 | 50.8% | 438 | 55.7 (52.2–59.2) | 3084 | 50.1 (48.9–51.4) | **0.003** |
| Male | 3414 | 49.2% | 348 | 44.3 (40.8–47.8) | 3066 | 49.9 (48.6–51.1) | |
| **Highest level of education completed** | | | | | | | |
| Illiterate | 38 | 0.9% | 3 | 0.6 (0.2–1.6) | 35 | 0.9 (0.7–1.3) | 0.083 |
| Elementary School | 1861 | 43.9% | 191 | 38.8 (34.6–43.2) | 1670 | 44.5 (43–46.1) | |
| High School | 1918 | 45.2% | 245 | 49.8 (45.4–54.2) | 1673 | 44.6 (43–46.2) | |
| College | 424 | 10.0% | 53 | 10.8 (8.3–13.7) | 371 | 9.9 (9–10.9) | |
| **Age** | | | | | | | |
| Mean (±SD) | 32.5 (±17.9) | | 32.7 (±13.6) | | 32.5 (±18.4) | | 0.249 |

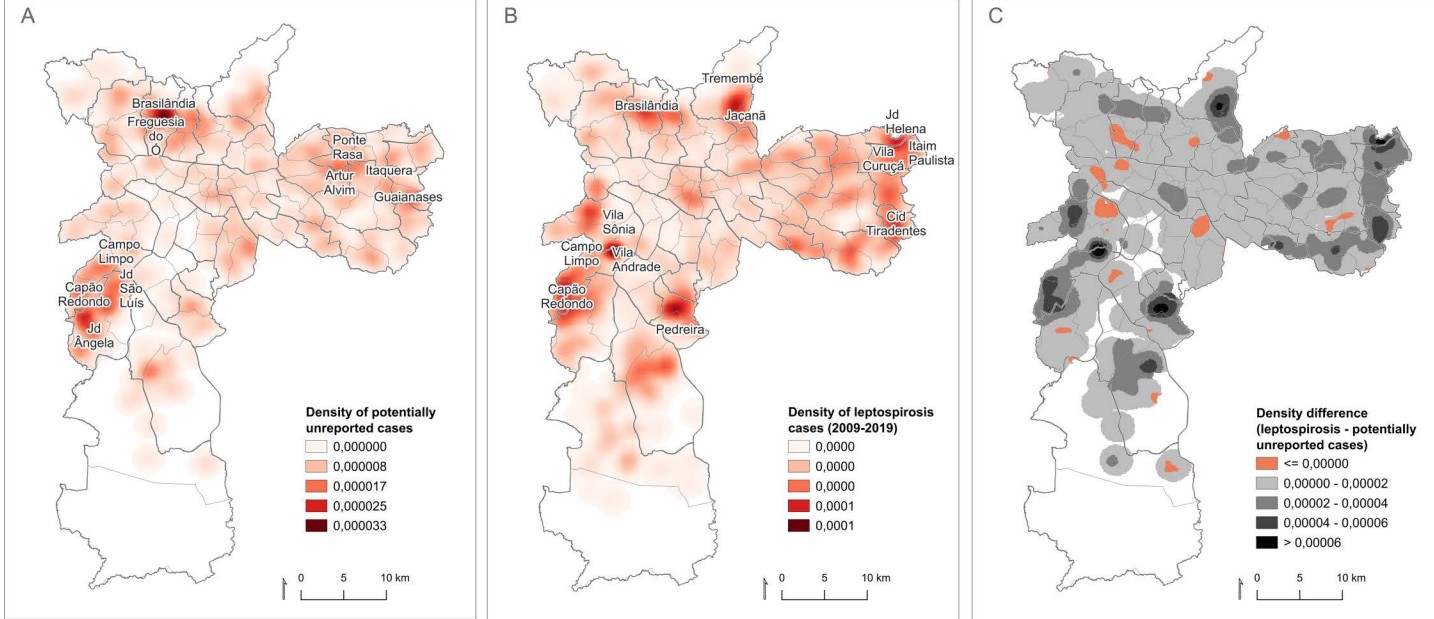

**Fig 2. Kernel density estimate of potentially unreported cases (A); Kernel density estimate of officially reported leptospirosis cases (B); and density difference map showing the kernel estimate of official cases with the density of unreported cases subtracted (C). The maps were built using the free and open source QGIS software (https://www.qgis.org/en/site/) based on shapefiles obtained from Instituto Brasileiro de Geografia e Estatística (IBGE - https://www.ibge.gov.br/geociencias/organizacao-do-territorio/malhas-territoriais/26565-malhas-de-setores-censitarios-divisoes-intramunicipais.html).**

The approximate location of residence was recovered for 6,044 patients tested (690 cases and 5,354 controls). The distribution of the 690 cases can be visualized in Fig 1A.

After including georeferenced data from each patient and spatially joining them with the selected spatial attributes (urban or rural area, slum areas, slums undergoing the urbanization process, irregular settlements, forest conservation units, and areas at risk of flooding), the association tests for the frequency of seropositivity outcome were unable to find differences in the distribution between cases and controls (S1 Table). Significant statistical differences were detected in the IDHM comparisons between UDHs of cases and controls (Table 2),

**Table 2.  IDHM differences between UDHs of cases and controls.**

|  | N valid | Missing | Mean (±SD) | Median (IIQ) | Min-Max | p-value |
|---|---|---|---|---|---|---|
| **Municipal Human Development Indexes (IDHM)** | | | | | | |
| Negative | 5372 | 2 | 0.761 (±0.07) | 0.757 (0.703–0.804) | 0.625–0.965 | **0.048** |
| Positive | 692 | 0 | 0.767 (±0.08) | 0.760 (0.710–0.8155) | 0.625–0.952 | |
| **Municipal Human Development Indexes - Education** | | | | | | |
| Negative | 5372 | 2 | 0.700 (±0.08) | 0.702 (0.645–0.752) | 0.516–0.948 | 0.119 |
| Positive | 692 | 0 | 0.705 (±0.08) | 0.704 (0.645–0.755) | 0.516–0.918 | |
| **Municipal Human Development Indexes - Longevity** | | | | | | |
| Negative | 5372 | 2 | 0.848 (±0.05) | 0.851 (0.806–0.896) | 0.737–0.957 | **0.032** |
| Positive | 692 | 0 | 0.853 (±0.06) | 0.854 (0.811–0.906) | 0.737–0.957 | |
| **Municipal Human Development Indexes - Income** | | | | | | |
| Negative | 5372 | 2 | 0.743 (±0.09) | 0.725 (0.678–0.774) | 0.618–1 | **0.028** |
| Positive | 692 | 0 | 0.751 (±0.09) | 0.729 (0.683–0.794) | 0.618–1 | |

including the general IDHM (p = 0.048) and its dimensions IDHM-Longevity (p = 0.032) and IDHM-Income (p = 0.028).

Regarding the density of potentially underreported cases, a high density was detected in specific locations in the North and South regions (Fig 2A). High-density clusters in the North region included the districts of Freguesia do Ó and Brasilândia, while the South region showed higher density in the districts of Campo Limpo, Jardim Ângela, Capão Redondo, and Jardim São Luís. Although less intense, high-density areas were also identified in the Eastern region, including the districts of Artur Alvim, Itaquera, Ponte Rasa, and Guaianases.

A map (Fig 1B) showing the density differences between cases and controls was constructed to determine whether these clusters represented an actual risk for potential underdiagnosis or were simply a result of high population density. This revealed that, with the exception of four relatively small areas, the clusters initially identified could not represent an increased risk of underreporting.

Spatial data from potentially underreported cases were further compared with the official leptospirosis records from the SINAN database. Between 2009 and 2019, 2,979 cases of leptospirosis were officially registered in São Paulo. Most of these patients were male (82.8%; 95% CI: 81.4%–84.1%), with a mean age of 39 years (SD ± 16 years). The distribution pattern of these cases showed that the highest density cluster was found in the administrative districts of Vila Andrade and Vila Sônia (Fig 2B), more specifically in Paraisópolis, which is the largest slum in São Paulo and that occupies part of both districts. Other spatial clusters were identified, including the districts of Pedreira, Campo Limpo, and Capão Redondo in the Southern region; Jaçanã, Tremembé, and Brasilândia in the Northern region; and Jardim Helena, Itaim Paulista, Vila Curuçá, and Cidade Tiradentes in the Eastern region. The highest densities were predominantly found in peripheral areas, mostly within slum communities.

By calculating the difference between the normalized density estimates of official diagnoses and potential underreported cases, it was found that the density of potentially underreported cases exceeds that of officially reported cases in 18 areas across the territory (Fig 2C). These areas span 28 administrative districts, including 11 in the South region, nine in the West region, and four each in the East and North regions. Only 12 public primary care facilities are located within these high-density underdiagnosis regions.

To define high-risk areas for potential underdiagnosis (Fig 3A), the Relative Risk (RR) was calculated, including population data per UDH and the number of cases and controls for each of the UDHs in the municipality of São Paulo. Eight high-risk clusters were identified, of

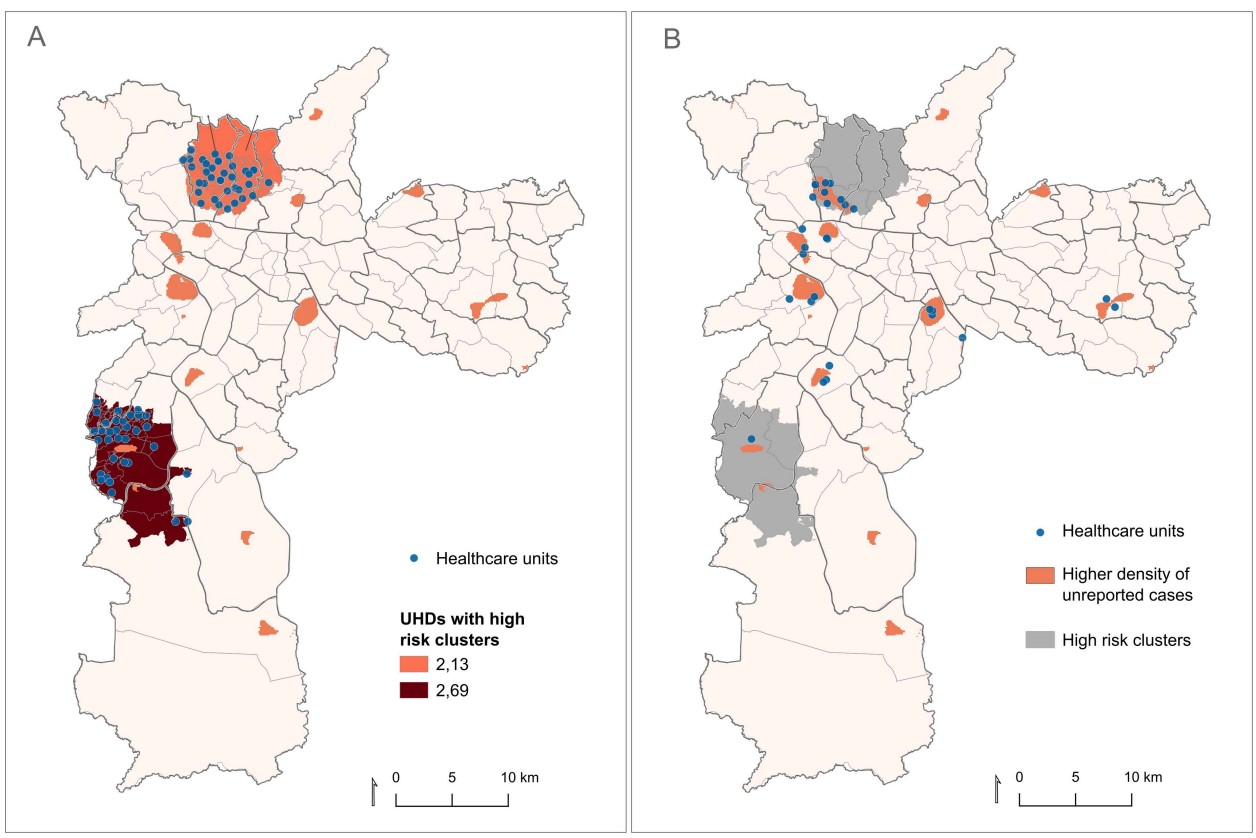

**Fig 3.** **(A) Map of the city of São Paulo showing UHDs within high-risk clusters for potentially unreported cases of leptospirosis, with healthcare units overlaid on the risk areas, and (B) areas with higher density of underdiagnosis compared to officially reported cases in SINAN, overlaid with risk clusters, including healthcare units located or within 500 meters of the highest density areas. The maps were built using the free and open source QGIS software (https://www.qgis.org/en/site/) based on shapefiles obtained from Instituto Brasileiro de Geografia e Estatística (IBGE - https://www.ibge.gov.br/geociencias/organizacao-do-territorio/malhas-territoriais/26565-malhas-de-setores-censitarios-divisoes-intramunicipais.html).**

which only two had statistical significance (Fig 3A): the first having an RR of 2.13 (p < 0.001) in the North Zone, covering 95 UDHs, and the second cluster with an RR of 2.69 (p < 0.001) in the South Zone of São Paulo, covering 67 UDHs. These areas include a total of 72 healthcare units, with 37 located in the North Zone and 35 in the South Zone of São Paulo.

Within the high-risk areas, three locations exhibited a higher density of potentially underreported cases compared to officially reported cases (Fig 3B). A total of 28 healthcare units were either overlaid or located within 500 meters of these high-density underdiagnosis areas for leptospirosis. Of these, ten were situated in the North Zone, directly within the high-risk areas, and one located in the South Zone.

## Discussion

This study stands as the first large-scale investigation to detect probable underrecognized cases of leptospirosis among dengue-negative patients with further spatial analysis and identification of risk areas. The proportion of seroreactive patients (11.3%) is consistent with other studies investigating leptospirosis among dengue-negative subjects [28,46,47], revealing that leptospirosis is also underreported in São Paulo. As the samples were collected between May and December, a period marked by low rainfall and a reduced incidence of leptospirosis

in São Paulo [48], the occurrence of unrecognized leptospirosis is likely to be even higher throughout the entire year. This is especially true given that disease transmission typically peaks during the summer season in urban environments [9].

This estimate, nevertheless, is markedly different from those officially reported by the SINAN. During the same period that samples were collected, only 64 cases of leptospirosis were officially recorded in São Paulo [36], corresponding to an incidence rate of 0.57 cases per 100,000 inhabitants. If all 786 seroreactive patients found in this study were considered to be confirmed cases and subsequently added to the official registries, the adjusted incidence of leptospirosis for the study period would be 13 times higher (7.56 cases per 100,000 inhabitants). Interestingly, this estimate is comparable to the incidence of HIV and significantly higher than that of other neglected diseases of public health importance in Brazil, such as leishmaniasis, chikungunya, schistosomiasis, Hansen's disease, Chagas disease, and yellow fever [36].

These results should be interpreted cautiously, though. Firstly, a positive IgM detection in a single serum sample is usually insufficient to confirm leptospirosis cases in Brazil [49], unless clinical and epidemiological evidence for infection is also present, such as impaired liver, renal or vascular function, or recent exposure to potential risk factors attributed to leptospiral transmission [22]. Since the information provided on registration forms for suspected dengue cases differs substantially from that collected for suspected leptospirosis cases, critical epidemiological information, as well as clinical and laboratory data needed to confirm infection in the seropositive ELISA patients were mostly missing. Secondly, although ELISA assays are widely used as a diagnostic tool for detecting leptospiral infection, playing a crucial role in disease surveillance, highly variable diagnostic sensitivity and specificity values for ELISA IgM testing have been described [50]. Two systematic reviews with meta-analysis, however, have shown encouraging results. The first, performed in 2012 by Signorini *et al.* [31], found a pooled diagnostic sensitivity and specificity of 77.9% and 91.3%, respectively. The second, performed by *Rosa et al.* (2017), revealed that IgM detection during the acute leptospiremic phase (between 3–7 days) in patients presenting with nonspecific symptoms such as myalgia and headache, had diagnostic sensitivity and specificity of 90.0% and 91.0%, respectively [51].

Additionally, false-positive results may influence overall seropositivity, primarily because IgM-based tests can indicate not only active disease, but also pre-existing immunity acquired prior to clinical presentation, particularly in populations with frequent exposure to the pathogen. Recent evidence has also shown that IgM antibodies can also be detected by ELISA tests in mice experimentally infected with saprophytic *Leptospira* [52]. Given the indigenous presence of these microorganisms in tropical environments, such interference in ELISA tests used for the diagnosis of human leptospirosis should be further explored.

It is also important to highlight that the ELISA results differed substantially from those obtained with MAT. Since the study population primarily consisted of patients assessed between 6 and 10 days after the onset of nonspecific febrile symptoms — a period corresponding to the early stages of anti-Leptospira antibody formation in patients truly infected with leptospires — this divergence was expected, once the ELISA testes are widely recognized as being more effective for detecting earlier infection than other serological techniques, including the MAT [1].

Overall, our results indicate that leptospirosis might be highly underrecognized in São Paulo and should not be overlooked as a major cause of febrile illness among dengue-negative patients. The high level of underrecognition of leptospirosis, as the one found in the present study, remains one of the main drivers of its neglected status, leading to imprecise or even misleading burden estimates and epidemiological planning by health authorities [14]. Even in the face of its clearly underestimated burden, official records indicate that leptospirosis

remains a major health threat in Brazil. The disease's impact has been recently quantified at 30.97 disability-adjusted life-years (DALYs) per 100,000 inhabitants in Brazil [8] — comparable to the burden caused by dengue and schistosomiasis, and higher than malaria and leishmaniasis [53], which are diseases that receive substantial research funding and significant resources for diagnostic and prevention programs in the country [54].

Despite these figures, leptospirosis still does not feature among the diseases prioritized by Brazil's Ministry of Health, as outlined in the National Agenda of Priorities in Health Research and its successive updates [54], a governmental consensus that is responsible for choosing which health conditions should receive greater research funding or incentives for improving prevention and control policies. More recently, in 2024, the Brazilian Ministry of Health has launched a national program for the eradication of socially determined diseases, called *Healthy Brazil* (Brasil Saudável), which also makes no mention of leptospirosis as a neglected or socially determined disease, sidelining any specific strategies or goals for preventing this illness [55].

This scenario suggests that initiatives to prevent leptospirosis may receive minimal political and legal support, which might contribute to the general public's lack of awareness about the disease and helps keep it largely marginalized from the public agenda, as previously suggested [56,57]. This does not imply, however, that efforts to prevent the disease are absent within the various levels of governance in the country. The Brazilian Ministry of Health has a permanent task force for rodent control, responsible for planning leptospirosis prevention actions and recommending specific actions in priority areas. In São Paulo, for example, the Leptospirosis and Rodent Surveillance and Control Program has been in effect since 1983, with recent revisions in 2006 and 2013 [58]. This initiative, primarily based on the identification of risk areas and the subsequent analysis of the socioeconomic, sanitation, and hydrological conditions in regions with a high incidence of the disease, has allowed the designation of zones eligible for specific preventive measures (called Áreas Programas), where actions such as rodent control and risk communication to primary healthcare units are implemented. Additionally, periodic cycles of intervention are put into practice to control leptospirosis in São Paulo.

The large number of seroreactive patients identified in this study allowed for a more detailed investigation of spatial patterns across the territory and a comparison with the distribution of officially registered cases in SINAN from 2009 to 2019, including the Program Areas established by local authorities. Since LABZOO is responsible for processing suspected dengue samples from all public municipal hospitals and outpatient medical assistance units, basic healthcare units, and emergency care units located within the city, as well as some hospitals from the state network (with a minimum portion sent to other public reference centers), the risk estimates tend to be representative for all municipality regions and subregions.

Our results show that potential underreported cases were clustered in several locations across the territory, most of which coincide with areas where SINAN cases were registered between 2009 and 2019. The comparison with SINAN cases revealed that in the areas containing the two main hotspots for official cases, Paraisópolis and Cidade Ademar, leptospirosis was rarely underdiagnosed, likely due to the high awareness of local health services about the disease. These results indicate that, despite the presence of mild and unrecognized cases, surveillance based on official records remains overall effective in identifying the main risk areas in the city. In this context, confirmed cases of leptospirosis may act as sentinel events, helping to define priority areas for prevention and rodent control. The analysis also showed, however, that several areas outside the official hotspots had proportionally more underreported cases than reported cases, with a total of 28 health services located within or near these areas, possibly indicating a lack of awareness about the disease. Risk assessment further revealed two major regions with high risk of underreporting. Within these areas, 72 healthcare units were

found, and strategic instruction of local workforce should focus on these areas, especially in the six healthcare units located in regions with a higher density of potentially underreported cases compared to officially reported cases.

The identification of areas with higher potential for underdiagnosis as carried out here may assist local health authorities in implementing strategies aiming to reduce the under-recognition of leptospirosis. These strategies should include educational campaigns targeting local populations, field training for family medicine professionals on the importance and risk factors of the disease, as well as raising awareness among primary care teams about the high number of underdiagnosed leptospirosis cases. Additionally, providing instructions to local healthcare workers on clinical and laboratory differentiation between leptospirosis and dengue based on previous studies that have identified comprehensive screening schemes and algorithms [59, 60] could be highly beneficial. Also, the distribution and use of point-of-care rapid tests in strategic primary care services could facilitate prompt and appropriate care, thereby increasing the chances for the early diagnosis and lethality reduction. Such initiatives could be particularly relevant after the recent introduction of the dengue vaccine into the Brazilian national immunization plan, which is expected to reduce the number of dengue cases. This reduction could, in turn, increase the proportion of febrile illnesses caused by leptospiral infection among dengue-negative patients presenting at local health services.

Finally, the demographic profile analysis of seroreactive individuals found that women were more likely to have underreported leptospirosis, which contrasts with the official global and local estimates that shows that up to 80% of official leptospirosis cases are registered in males [16,61]. These findings suggest that while males are more likely to present severe cases and, therefore, be more easily recognized as leptospirosis cases, women, who may be more inclined to seek care for oligosymptomatic or mild cases, are consequently more likely to be underdiagnosed. This demographic profile may help guide local authorities in providing specific instructions to healthcare units with a higher likelihood of underreporting.

Regarding socio-environmental attributes, no significant differences were found in the frequency of seroreactivity in patients living within or outside slum areas, slums undergoing urbanization process, irregular settlements, or areas at risk of flooding, as opposed to literature evidence showing that leptospirosis is more frequently seen in populations living in urban disadvantaged communities [12]. This finding indicates that that under-recognition may affect different social strata indistinctly. Yet, differences in residence location IDHM between cases and controls were detected in our study, where addresses within UDHs with higher IDHM levels being more frequent in the cases subgroup than in the controls. This surprising result may be explained by the fact that, in São Paulo, social disparities are strikingly pronounced, and even within the same administrative district or an even smaller territorial division, such as UDHs, significant social differences can still be found, thus indicating that increased risk may be assessed only at an individual level. Our findings may suggest that poor individuals living in higher-income UDHs might be more likely to be underdiagnosed, mostly because healthcare professionals working in these locations might consider that local leptospiral transmission in unlikely.

This study has several limitations, including the use of secondary databases, which might contain missing data or errors in patient record entries. This study was further limited by the inability to reevaluate seropositive patients to obtain convalescent samples for confirming leptospiral infection, either through MAT or other diagnostic techniques. Additionally, restricted access to patients' clinical data hindered the confirmation of infection using clinical epidemiological criteria.

Possible false-positive results in the ELISA could have been minimized by applying more stringent thresholds than the one recommended by the manufacturer, as recommended

elsewhere [50]. False-negative results, on the other hand, are unlikely, given that most patients had samples taken within six days of symptom onset, largely because IgM testing for dengue is conducted only after symptoms persist beyond six days. The incorporation of more advanced approaches, such as molecular detection techniques or serum sample culture methods, should be considered by local authorities as promising strategies to achieve more precise and dependable confirmation of leptospirosis in future diagnostic protocols.

Finally, the study did not include samples taken during the summer season, when leptospiral transmission is expected to be higher due to increased precipitation, flooding, and extreme weather events. These conditions typically increase disease awareness among healthcare providers, which may influence the degree of underrecognition. The selected time frame has consequently limited any conclusions about temporal patterns or the seasonality of underreporting cases. Also, the study included only samples of patients treated in the municipality's public network, therefore excluding samples of patients covered by healthcare plans or out-of-pocket expenditures. This limitation is mitigated by the fact that most citizens are served by public services in São Paulo, [62] especially those living in conditions at risk for the disease.

## Conclusion

To the best of our knowledge, this study represents the first systematic effort to identify hotspots and risk areas for the occurrence of unrecognized cases of leptospirosis. Through spatial analysis, we successfully identified multiple areas with a higher risk and occurrence of potentially undetected cases. These findings provide a critical foundation for local health authorities to design and implement targeted strategic interventions aimed at reducing the underrecognition of leptospirosis. By pinpointing these high-risk areas, public health strategies can be more effectively focused on enhancing disease surveillance, improving diagnostic capacities, and conducting timely community outreach and education.

Future research efforts should aim to build upon the methods described in this study by incorporating data from an expanded period, which would allow for the identification of temporal trends and long-term patterns in leptospirosis transmission. Additionally, the integration of more comprehensive databases, including environmental and socioeconomic variables, could provide a deeper understanding of the factors contributing to the transmission and underreporting of leptospirosis. The use of artificial intelligence and machine learning algorithms could further enhance surveillance efforts by enabling the rapid analysis of complex datasets, facilitating more timely and accurate responses to the underrecognition of the disease or even emerging outbreaks.

## Supporting information

**S1 Table. Distribution of Positive and Negative Patients by Spatial Attributes and Association Test Results.** The table summarizes the number (N) and percentage (%) of patients classified as positive or negative with spatial attributes including areas at risk of flooding, forest conservation units, slum areas, irregular settlements, slums undergoing urbanization, and urban versus rural locations. The results include p-values from chi-square tests to assess the association between patient status and spatial attributes, with confidence intervals (95% CI) provided for percentage estimates.
(DOCX)

**S2 Table. Raw data related to the study.** This table includes the raw data used in the analyses of the study, limited to the data permissible for sharing in compliance with the Brazilian General Data Protection Law (Lei Geral de Proteção de Dados - LGPD).
(XLSX)

## Acknowledgments

We would like to thank the Bacterial Zoonoses Laboratory of the Department of Preventive Veterinary Medicine and Animal Health, especially Denise Batista Nogueira and Gisele Oliveira de Souza, for their valuable support throughout this study. We also extend our gratitude to the Brazilian Ministry of Health for providing the ELISA kits used in this research. We also would like to thank Felipe Fornazari for his valuable contribution.

## Author contributions

**Conceptualization:** Stephanie Bergmann Esteves, Aline Gil Alves Guilloux, Eduardo de Masi, Marcos Bryan Heinemann, Bruno Alonso Miotto.

**Data curation:** Stephanie Bergmann Esteves, Luciano Marcondes de Oliveira, Isabelle Martins Ribeiro Ferreira, Marcos Bryan Heinemann, Bruno Alonso Miotto.

**Formal analysis:** Stephanie Bergmann Esteves, Aline Gil Alves Guilloux, Evelyn Moura de Lima, Gabriele Yumi Ramalho, Priscila de Castilho Luna.

**Investigation:** Stephanie Bergmann Esteves, Adriana Cortez, Eduardo de Masi, Marcos Bryan Heinemann, Bruno Alonso Miotto.

**Methodology:** Stephanie Bergmann Esteves, Luciano Marcondes de Oliveira, Aline Gil Alves Guilloux, Adriana Cortez, Marcos Bryan Heinemann, Bruno Alonso Miotto.

**Project administration:** Stephanie Bergmann Esteves, Luciano Marcondes de Oliveira, Eduardo de Masi, Isabelle Martins Ribeiro Ferreira, Marcos Bryan Heinemann, Bruno Alonso Miotto.

**Resources:** Marcos Bryan Heinemann, Adriana Cortez, Jose Soares Ferreira Neto, Bruno Alonso Miotto.

**Software:** Stephanie Bergmann Esteves, Aline Gil Alves Guilloux.

**Supervision:** Marcos Bryan Heinemann, Bruno Alonso Miotto.

**Visualization:** Bruno Alonso Miotto.

**Writing – original draft:** Stephanie Bergmann Esteves, Luciano Marcondes de Oliveira, Adriana Cortez, Eduardo de Masi, Isabelle Martins Ribeiro Ferreira, Evelyn Moura de Lima, Gabriele Yumi Ramalho, Priscila de Castilho Luna, Jose Soares Ferreira Neto, Marcos Bryan Heinemann, Bruno Alonso Miotto.

**Writing – review & editing:** Stephanie Bergmann Esteves, Luciano Marcondes de Oliveira, Aline Gil Alves Guilloux, Adriana Cortez, Eduardo de Masi, Isabelle Martins Ribeiro Ferreira, Evelyn Moura de Lima, Gabriele Yumi Ramalho, Priscila de Castilho Luna, Jose Soares Ferreira Neto, Marcos Bryan Heinemann, Bruno Alonso Miotto.

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
