## [Decision Letter · Decision Letter 0]

8 Jan 2025

PNTD-D-24-01662

Into the spotlight: a spatial study of potentially underreported leptospirosis among dengue-negative patients in São Paulo city, Brazil

Dear Dr. Esteves,

Thank you for submitting your manuscript to PLOS Neglected Tropical Diseases. After careful consideration, we feel that it has merit but does not fully meet PLOS Neglected Tropical Diseases's publication criteria as it currently stands. Therefore, we invite you to submit a revised version of the manuscript that addresses the points raised during the review process. Specifically, please present MAT results, which will provide insights on the types of infecting leptospires. If not possible for all samples, then at least 10 will be useful information.

Please submit your revised manuscript within 60 days Mar 09 2025 11:59PM. If you will need more time than this to complete your revisions, please reply to this message or contact the journal office at plosntds@plos.org. Please include the following items when submitting your revised manuscript:

We look forward to receiving your revised manuscript.

Kind regards,

Brian Stevenson, Ph.D.

Academic Editor

Justin Remais

Section Editor

Shaden Kamhawi

co-Editor-in-Chief

Paul Brindley

co-Editor-in-Chief

**Journal Requirements:**

At this stage, the following Authors/Authors require contributions: Stephanie Bergmann Esteves, Luciano Marcondes de Oliveira, Aline Gil Alves Guilloux, Adriana Cortez, Eduardo de Masi, Isabelle Martins Ribeiro Ferreira, Evelyn Moura de Lima, Gabriele Yumi Ramalho, Priscila de Castilho Luna, Jose Soares Ferreira Neto, Marcos Bryan Heinemann, and Bruno Alonso Miotto. Please ensure that the full contributions of each author are acknowledged in the "Add/Edit/Remove Authors" section of our submission form.

- ® on pages: 7 line 221, and 8 line 231.

3) Tables should not be uploaded as individual files. Please remove these files and include the Tables in your manuscript file as editable, cell-based objects. For more information about how to format tables, see our guidelines:

https://journals.plos.org/plosntds/s/tables 

Potential Copyright Issues:

i) Figures 1, 2, and 3. Please (a) provide a direct link to the base layer of the map (i.e., the country or region border shape) and ensure this is also included in the figure legend; and (b) provide a link to the terms of use / license information for the base layer image or shapefile. We cannot publish proprietary or copyrighted maps (e.g. Google Maps, Mapquest) and the terms of use for your map base layer must be compatible with our CC BY 4.0 license.

6) We note that your Data Availability Statement is currently as follows: "All relevant data are within the manuscript and its Supporting Information files." Please confirm at this time whether or not your submission contains all raw data required to replicate the results of your study. Authors must share the “minimal data set” for their submission. PLOS defines the minimal data set to consist of the data required to replicate all study findings reported in the article, as well as related metadata and methods (https://journals.plos.org/plosone/s/data-availability#loc-minimal-data-set-definition).

7) Your current Financial Disclosure states, "This study received funding support in the form of fellowships granted to the authors. SBE was supported by the São Paulo Research Foundation (FAPESP, grant number 2021/02534-9, https://fapesp.br/). MBH was supported by the National Council for Scientific and Technological Development (CNPq, grant number 310462/2021, https://www.gov.br/cnpq/). The funders had no role in study design, data collection and analysis, decision to publish, or preparation of the manuscript. "

However, your funding information on the submission form indicates receiving no funds. Please ensure that the funders and grant numbers match between the Financial Disclosure field and the Funding Information tab in your submission form. Note that the funders must be provided in the same order in both places as well.

Please indicate by return email the full and correct funding information for your study and confirm the order in which funding contributions should appear. Please be sure to indicate whether the funders played any role in the study design, data collection and analysis, decision to publish, or preparation of the manuscript.

**Reviewers' Comments:**

Reviewer's Responses to Questions

**Key Review Criteria Required for Acceptance?**

**Methods**

-Are the objectives of the study clearly articulated with a clear testable hypothesis stated?

-Is the study design appropriate to address the stated objectives?

-Is the population clearly described and appropriate for the hypothesis being tested?

-Is the sample size sufficient to ensure adequate power to address the hypothesis being tested?

-Were correct statistical analysis used to support conclusions?

-Are there concerns about ethical or regulatory requirements being met?

Reviewer #1: The objective was clearly articulated and the study design was proper. The method section needs a minor revision to provide the details of the statistical test conducted. I have added few recommendation to improve the discussion section.

Reviewer #2: The ELISA method utilized in this study may be suitable for use in tertiary clinics and certain specific scenarios. However, for a cross-sectional study of this nature, the MAT would have been more appropriate. Given the number of samples available to the authors, not employing MAT is a missed opportunity, as it offers significantly more comprehensive information regarding the extent of infection, as well as the serovars or serogroups involved.

**Results**

-Does the analysis presented match the analysis plan?

-Are the results clearly and completely presented?

-Are the figures (Tables, Images) of sufficient quality for clarity?

Reviewer #1: All results were presented clearly and completely.

Reviewer #2: ELISA results and other analyses are clearly presented.

**Conclusions**

-Are the conclusions supported by the data presented?

-Are the limitations of analysis clearly described?

-Do the authors discuss how these data can be helpful to advance our understanding of the topic under study?

-Is public health relevance addressed?

Reviewer #1: The conclusion needs a minor modification and authors are requested to address the comments included in the summary and general comments section.

Reviewer #2: Conclusion drawn are justified.

**Editorial and Data Presentation Modifications?**

Reviewer #1: Data were presented well and authors are requested to follow the next section for recommendations for revising the manuscript.

Reviewer #2: (No Response)

**Summary and General Comments**

Reviewer #1: Esteves and the team conducted a spatial analysis to identify potentially underreported cases of leptospirosis among dengue-negative patients in São Paulo City, Brazil. The study analyzed 6,936 serum samples from patients who tested negative for dengue at public health services across São Paulo. These samples were subsequently tested for anti-leptospiral IgM serology to identify possible unreported leptospirosis cases. Results revealed that 786 (11.3%) samples were seroreactive for leptospirosis, with a higher prevalence observed in women than in men.

The spatial analysis identified 18 high-risk clusters for potentially undiagnosed leptospirosis, primarily located in peripheral areas and two prominent clusters in the northern and southern regions of the city. This investigation marks the first large-scale study highlighting potential hotspots and high-risk zones for unrecognized leptospirosis in São Paulo, a city that records approximately 4,000 reported cases annually.

While the study acknowledges certain limitations, the findings offer a critical foundation for designing and implementing targeted interventions to address the underdiagnosis of leptospirosis. This research holds significant public health implications and has the potential to draw the attention of Brazil's Ministry of Health. By addressing high-risk and underreported regions, such efforts could significantly contribute to the country's broader national health strategies, particularly as leptospirosis incidents rise globally, including in Brazil.

Few questions and recommendations are enlisted below:

1. This study successfully identified 11.3% of unrecognized leptospirosis cases from 6,937 dengue-negative serum samples collected from patients with febrile illnesses. What other potential illnesses could present the rest of the ~89% with symptoms similar to dengue or leptospirosis?

2. Line 332: Please include the detais of statistical test that were conducted.

1. What is the likelihood of detecting anti-leptospiral IgM positivity in serum from a patient exposed to saprophytic Leptospira, which are indigenous in nature, alongside pathogenic Leptospira? Authors are encouraged to reference studies (PMID: 39527098, 34249775, 35392072, 39018329) to emphasize the significance of their indigenous presence, evolutionary dynamics in nature, and their varied clinical outcomes that are similar to dengue or other diseases.

3. Line 324-328: What is the probability of identifying a seropositive cluster with saprophytic Leptospira infection, which, while not posing a direct threat to humans, could still induce seropositivity to anti-leptospiral IgM? Authors are requested to add that in the discussion around Line 467-471.

4. The authors are encouraged to discuss the presence of Leptospira in other clinical samples, such as genital specimens, to provide insights into additional transmission routes (e.g., PMID: 35446113). This discussion could aid future studies exploring the potential for sexual transmission and help determine whether there are any sex-based differences in leptospiral infection dynamics.

5. Line 600-612: The authors have outlined the limitations of the study. It is recommended to include a section proposing more stringent diagnostic approaches to address the issues of misdiagnosis or underdiagnosis of leptospirosis. Incorporating advanced methods, such as molecular detection techniques or culture methods from serum samples, should be highlighted as potential strategies for more accurate and reliable confirmation of leptospirosis in future diagnostic protocols.

Reviewer #2: This cross-sectional study by Bergmann Esteves and colleagues analyzed febrile, dengue-negative patients in São Paulo city for leptospiral antibodies using a commercially available ELISA. The results indicated that 11.3% of the tested patients were seropositive, and spatial analysis identified 18 high-risk clusters of potentially unrecognized leptospirosis in the city.

A key concern is the choice of assay for the serosurvey. While the ELISA method used in this study may be appropriate for tertiary clinics and specific scenarios, the MAT would have been more suitable for a study of this nature. Given the substantial number of samples available to the authors, not employing MAT is a missed opportunity, as it is more specific and provides far more detailed information on the extent of infection and the specific serovars or serogroups involved.

PLOS authors have the option to publish the peer review history of their article (what does this mean? ). If published, this will include your full peer review and any attached files.

**Do you want your identity to be public for this peer review?** For information about this choice, including consent withdrawal, please see our Privacy Policy .

Reviewer #1: **Yes: ** Suman Kundu

Reviewer #2: No

**Figure resubmission:**
---

## [Editor Report · Decision Letter 1]

4 Feb 2025

Dear Ms Esteves,

We are pleased to inform you that your manuscript 'Into the spotlight: a spatial study of potentially underreported leptospirosis among dengue-negative patients in São Paulo city, Brazil' has been provisionally accepted for publication in PLOS Neglected Tropical Diseases.

Best regards,

Brian Stevenson, Ph.D.

Academic Editor

Justin Remais

Section Editor

Shaden Kamhawi

co-Editor-in-Chief

Paul Brindley

co-Editor-in-Chief

---

## [Editor Report · Acceptance letter]

Dear Ms Esteves,

We are delighted to inform you that your manuscript, "Into the spotlight: a spatial study of potentially underreported leptospirosis among dengue-negative patients in São Paulo city, Brazil," has been formally accepted for publication in PLOS Neglected Tropical Diseases.

Best regards,

Shaden Kamhawi

co-Editor-in-Chief

Paul Brindley

co-Editor-in-Chief
